# Molecular Evidence Supports Five Lineages within *Chiropotes* (Pitheciidae, Platyrrhini)

**DOI:** 10.3390/genes14071309

**Published:** 2023-06-21

**Authors:** Jeferson Carneiro, Iracilda Sampaio, José de S. e Silva-Júnior, Antonio Martins-Junior, Izeni Farias, Tomas Hrbek, Jean Boubli, Horacio Schneider

**Affiliations:** 1Institute of Coastal Studies, University Campus of Bragança, Federal University of Pará, Belém 66075-110, Pará, Brazil; ira@ufpa.br (I.S.); horacio@ufpa.br (H.S.); 2Museu Paraense Emílio Goeldi, Mastozoology, Belém 66077-830, Pará, Brazil; 3Federal Institute of Education, Science and Technology of Pará, Campus Tucuruí, Tucuruí 68455-210, Pará, Brazil; antonio_mgmartins@hotmail.com; 4Laboratório de Evolução e Genética Animal, Federal University of Amazonas, Manaus 69067-005, Amazonas, Brazil; izeni@evoamazon.net (I.F.); tomas@evoamazon.net (T.H.); 5School of Environment and Life Sciences, University of Salford, Salford M5 4WT, UK; j.p.boubli@salford.ac.uk

**Keywords:** cuxiu, phylogeny, biogeography *Chiropotes*

## Abstract

Pitheciines have unique dental specializations among New World monkeys that allow them to feed on fruits with hard pericarps, thus playing a major role as seed predators. The three extant pitheciine genera, *Pithecia*, *Cacajao* and *Chiropotes*, are all endemic to the Amazon region. Because of the uncertainties about interspecific relationships, we reviewed the systematics and taxonomy of the genus *Chiropotes*. The phylogenetic analyses were performed based on Maximum Likelihood and Bayesian Inference, while species delimitation analyses were carried out using multispecies coalescent methods. In addition, we estimated genetic distances, divergence time and the probable ancestral distribution of this genus. Our results support five species of *Chiropotes* that emerged during the Plio-Pleistocene. Biogeographic estimates suggest that the ancestor of the current *Chiropotes* species occupied the endemism areas from Rondônia and Tapajós. Later, subsequent radiation and founder effects associated with the formation of the Amazonian basins probably determined the speciation events within *Chiropotes*.

## 1. Introduction

The common English name for the genus *Chiropotes* Lesson, 1840, is bearded saki. Nonetheless, Cuxiu (regional popular name) has been proposed as the most suitable designation because of cultural and historical reasons [1]. *Chiropotes* are New World primates that, together with *Cacajao* Lesson, 1840, and *Pithecia* Desmarest, 1804, make up the subfamily Pitheciinae. These three genera are characterized by their specialized dentition, which assures the efficient processing of hard seeds and fruits which have a hard pericarp [2]. This specialization is more pronounced in *Chiropotes* and *Cacajao*, since they have more robust skulls and jaws than *Pithecia* [3]. *Chiropotes* are medium-sized monkeys that form social groups of 10–30 members, composed of multiple adult males and females and their offspring. These primates have a fission–fusion social system, in which the principal group divides frequently into smaller subgroups, depending on the distribution of resources before further reconnection [4]. Typically, *Chiropotes* inhabit the middle and upper levels of the forest canopy, where they move rapidly and agilely [3].

The genus *Chiropotes* was originally divided into three species: *Chiropotes satanas* (Hoffmannsegg, 1807), *Chiropotes chiropotes* (Humboldt, 1811) and *Chiropotes albinasus* (I. Geoffroy and Deville, 1848) [5]. However, Hershkovitz [6], based on morphological data, recognized only two species of cuxiu, *C. albinasus* and *C. satanas*, with the latter species being divided into three subspecies: *C. s. satanas*, *C. s. chiropotes* and *C. s. utahicki*. Silva Júnior and Figueiredo [7], based on a fragment of cytochrome b, suggested that the subspecies *C. s. satanas* and *C. s. utahicki* should be raised to species status. Additionally, they also proposed that populations from opposite banks of the Branco River were distinct species. Later, based on cytogenetic, morphological and mtDNA data, these two forms were named *Chiropotes israelita* Spix, 1823, and *C. chiropotes* [8].

Since *C. israelita* is a junior synonym of *C. chiropotes*, the names *C. israelita* and *C. chiropotes* are not valid to designate these putative possible species from Branco River margins, because it infringes the rule of priority of the International Code of Zoological Nomenclature (ICZN) (Silva Júnior unpublished). Thus, *C. chiropotes* and *Chiropotes sagulatus* (Traill, 1821) are proposed to designate the taxa that inhabit the west and east banks of the Branco River, respectively. Accordingly, this classification with five species of *Chiropotes* (*C. chiropotes*, *C. albinasus*, *C. sagulatus*, *C. satanas* and *C. utahicki*) was followed in the present study [6,7,8].

The International Union for Conservation of Nature (IUCN) considers *C. chiropotes* [9] and *C. sagulatus* [10] as Least Concern, while *C. albinasus* [11] and *C. utahicki* [12] are evaluated as Endangered and *C. satanas* [13] as Critically Endangered in red lists.

Primates play an important role in forest ecology, such as seed dispersal or seed predation [14,15]. However, due to human actions, many species of primates have faced remarkable population decline [16]. In these circumstances, taxonomic rigor is crucial for conservation strategies aimed at preventing their extinctions from becoming effective, which applies to the case of the cuxius.

The cuxius are endemic to Amazon region, where each species occupies an allopatric range separated from congeners by major rivers [6] (Figure 1). *Chiropotes satanas* is restricted to the eastern margin of the Tocantins-Araguaia basin, being found in the Brazilian states of Pará and Maranhão [8,17]. *Chiropotes utahicki* inhabits the Xingu–Tocantins interfluve, but the southern range limit remains uncertain [6]. *Chiropotes albinasus* is found between the Xingu and Madeira Rivers [6], as far south as the Guaporé River in Rondônia (Brazil), although this species is not recorded in most of the state of Rondônia between the Mamoré, Madeira and Ji-Paraná Rivers [18]. *Chiropotes chiropotes* occurs on the northern Amazon, from Venezuela to the west bank of the Branco River, while *C. sagulatus* is restricted from the east bank of the Branco River to the state of Amapá [6].

As mentioned before, changes in the taxonomy of *Chiropotes* have been proposed by DNA analyses [7,8], but they were based only on mtDNA markers. In the present study, we used a molecular approach to review the taxonomy of the cuxius, inferring their phylogenetic relationships, divergence times and ancestral areas of each clade by incorporating both nuclear and mitochondrial markers.

## 2. Materials and Methods

### 2.1. Sampling

Twenty samples of blood and muscle tissue preserved in alcohol were analyzed in the present study. These included four *Chiropotes* taxa, i.e., *C. albinasus* (*n* = 4 samples), *C. satanas* (*n* = 3), *C. utahicki* (*n* = 5) and *C. sagulatus* (*n* = 3), as well as *Cacajao melanocephalus* (*n* = 3), *Pithecia pithecia* (*n* = 1) and *Plecturocebus moloch* (*n* = 1). These samples were obtained from the DNA databanks at the Federal University of Pará (UFPA) in Belém and from the Federal University of Amazonas (UFAM) in Manaus, both in Brazil. The UFPA samples are referenced in previous studies [19,20,21]. Samples from UFAM were obtained from the wild under permits 40217-1 and 5135-1, supplied by the Instituto Chico Mendes de Conservação da Biodiversidade—ICMBio. The geographical origins of each sample are shown in Table 1. The specimens were identified based on external morphological characteristics sensu Hershkovitz [6].

Additional sequences of *C. chiropotes* from GenBank were also included in the analyses (for GenBank accession numbers, see Appendix A). These sequences are derived from samples collected by Bonvicino et al. [8] near the municipality of Barcelos, Manaus, Brazil, on the east bank of the Branco River. Bonvicino et al. [8] generated cytochrome b (Cyt B) sequences, while Perelman et al. [22] generated the sequences of 54 nuclear markers from the same samples. Thus, although these mitochondrial and nuclear sequences were generated in two different studies they were analyzed simultaneously in the present study because they derived from the same individual.

### 2.2. Laboratory Procedures

Total DNA was extracted from the samples using the Wizard Genomic kit (Promega Corporation, Madison, WI, USA), following the manufacturer’s instructions. Eight molecular markers were analyzed in the present study, including two mitochondrial genes (Cytochrome oxidase I—COI, and Cyt b) and five nuclear regions (ABCA1, AXIN, CHRNA1, DCTN2 and ERC2) (Appendix A). These markers were amplified via polymerase chain reaction (PCR) as follows: initial denaturation at 95 °C for 5 min, followed by 35 cycles of denaturation at 94 °C for 30 s, annealing for 45 s and extension at 72 °C for 45 s, plus a final extension of 5 min at 72 °C. Each PCR comprised approximately 50 ng of the genomic DNA, 2.4 μL of dNTPs (1.25 mM), 1.5 μL of 10X buffer solution (200 mM Tris-HCl, 500 mM KCl), 1 μL of MgCl 2 (25 mM), 1 μL of each primer (0.2 μM), 1 U of Taq DNA polymerase and ultrapure water to a final volume of 15 μL. The PCR products were purified using polyethylene glycol (PEG) 8000 and ethanol [23]. Sequence reactions were performed using the Big Dye sequencing kit v.3.1 (Life Technologies, Carlsbad, CA, USA), and sequencing was carried out in an ABI 3500XL (Life Technologies, Carlsbad, CA, USA) automatic sequencer.

### 2.3. Data Partitioning, Phylogenetic Analyses and Estimates of Genetic Distance

The sequences were aligned using ClustalW [24] and edited manually in BioEdit v.7.2.5 [25]. PartitionFinder v.1.1.0 [26] was used to identify the optimal data partitioning and evolutionary models. Exonic regions and mitochondrial genes were partitioned by codons, while introns and UTRs were partitioned by fragments. We used the greedy algorithm in PartitionFinder and the Bayesian Information Criterion (BIC) to choose the most suitable partitioning scheme for our database. The best partitioning scheme separated the data into two partitions: (I) the third base of both COI and Cyt B markers, and (II) the remaining markers. The TRN and HKY+G evolutionary models were selected for partitions I and II, respectively.

Phylogenetic analyses were based on Maximum Likelihood (ML) and Bayesian Inference (BI) approaches. The ML analysis was run in RAxML v.8 [27], and the most likely tree was identified by the majority consensus rule of 1000 bootstrap pseudo-replications. The Bayesian Inference was run in MrBayes v.3.2.6 [28], with two independent runs of four Monte Carlo Markov chains (MCMC), one cold and three heated, over 5,000,000 generations, with trees and parameters being sampled every 5000 generations. A 25% burn-in was established for these analyses (nruns = 2 nchains = 4 ngen = 1000.000 samplefreq = 5000; relburnin = yes burninfrac = 0.20).

The other two analyses were performed in MrBayes [28], separating mitochondrial from nuclear genes, and the evolutionary models were estimated using the software Kakusan v. 4.0 [29] (Appendix A). In both cases, the runs were checked for convergence by visualizing the log of posterior probability within and between the independent runs for each analysis, ensuring an average standard deviation of split frequencies below 0.005 and a potential scale reduction factor (PSRF) for estimated parameters close to 1.0. The “Effective Sample Size” (ESS) of all parameters was checked both in Tracer v.1.6 [30] and in MrBayes (“pstat” files), being regarded as adequate only when values were above 200.

Intraspecific and interspecific genetic distances using mitochondrial or nuclear markers or the whole database (mitochondrial and nuclear) were estimated in MEGA v 6.0 [31] using the Kimura 2-parameter (K2P) model [32].

### 2.4. Divergence Times, Species Tree, Species Delimitation and Biogeographic Analyses

Divergence time (DT) analyses were run on the multilocus sequence data in BEAST v.1.8.4 [33]. Proteropithecia, a pitheciine stem, dated from 15.7 million years ago (Ma) [34], was used as a calibration point (offset). A lognormal distribution was used (Mean = 1.17 and Stdev = 0.75) for the prior time of the most recent common ancestor (TMRCA). The uncorrelated relaxed clock [35] was the prior clock type, and the Yule process was the prior model for the tree [36]. Three independent runs were processed with 50 million generations and log parameters being registered at every 5000 generations. Logs were combined using LogCombiner v.1.8.3 [33] and trees summarized in TreeAnnotator v.1.8.4 [33]. A burn-in of 20% was used and the run was considered satisfactory when all ESS values checked in Tracer v.1.6 [30] were equal to or greater than 200.

Species were delimited using the software Bayesian Phylogenetics and Phylogeography (BPP) v.3.3a [37]. We tested the hypothesis of five potential species within *Chiropotes* (*C. albinasus*, *C. satanas*, *C. utahicki*, *C. chiropotes* and *C. sagulatus*) based on the ML and BI phylogenetic topologies (Figure 2).

The delimitation of species by the BPP is sensitive to the choice of the priors of population size (θ) and divergence time τ [38]. Therefore, in order to find the values for θ and τ, we carried out the analyses using the option A00 (speciesdelimitation = 0, speciestree = 0) with and without data. The priors obtained are similar to those used in the BPP default for species delimitation analyses (θ prior: α = 2; β = 2000 and τ prior α = 2; β = 25,000). A dirichlet prior α = 1 was used for all τ values. The run was set up with 500,000 generations, sampled at a frequency of 5, with burn-in of 50,000.

We also performed the analysis of species delimitation using “Species Tree and Classification Estimation, Yarely”—STACEY v.1.2.1 [39] in BEAST2 [40], which is an extension of the multispecies coalescence model used in * BEAST [41] that estimates the species tree based on a birth–death–collapse model. For this analysis, each marker represented a partition and the discrimination of species was based on the five main cuxius clades retrieved in ML and BI analyses. The input files (.xml) were created using BEAUti. The nucleotide substitution models were established in Kakusan v. 4.0 [29] (see Appendix A). As for other priors, the birth–death model was used to estimate the species tree (priors: Collapse Height = 0.001, Collapse Weight = 0.5 using a prior beta (1.1) around (0.1), Ploidy: equal to 2 for nuclear genes and 0.5 for mtDNA genes) and the Non-Correlated Lognormal model was selected to describe the relaxed molecular clock. The MCMC analysis was performed by 109 generations, sampled every 10,000 generations using 25% burn-in. The obtained logs were evaluated in Tracer to verify if satisfactory ESS values (≥200) were obtained.

In order to estimate the biogeographic history of distributional ranges of cuxius, we used the BioGeoBEARS “Biogeography with Bayesian (and likelihood) Evolutionary Analysis” R script [42]. We performed a total of 12 models implemented in BioGeoBEARS. The models include the Dispersal–Extinction Cladogenesis Model (DEC), a likelihood version of the Dispersal–Vicariance Analysis (DIVALIKE), and a version of the Bayesian Inference of historical biogeography for discrete areas (BAYAREALIKE), as well as Jump dispersal (+J) versions of these three models. The Jump dispersal (also referred to as founder–event speciation) corresponds to a scenario when a new population is founded from a colonization event and instantly becomes genetically isolated from the ancestral population [42]. Additionally, to account for the influence of geographic distance on dispersal, we also implemented distance-based models available in BioGeoBEARS (+X models; DEC+X, DEC+J+X, DIVALIKE+X, DIVALIKE+J+X, BAYAREALIKE+X, BAYAREALIKE+J+X). These models were included because the greater the geographic distance, the lower the dispersal probability becomes [43]. The distances among areas were estimated using Google Earth and Quantum GIS [44] based on the central point of each area, as defined by the intersection between the north–south and east–west boundaries.

Six discrete biogeographic areas were defined: Belém (BE), Xingu (XI), Tapajós (TA), Rondônia (RO), Guianas (GU) and Tepui (PA) (Appendix A). These areas reflect known centers of vertebrate endemism, as described by Ribas et al. [45], with the addition of Tepui [46]. The maximum possible number of ancestral states that could be assigned to each node was set to six, so that each taxon or node could occur simultaneously in all pre-defined areas. The tree of divergence time estimates was used to infer the ancestral range probabilities. The outgroups (*Plecturocebus*, *Pithecia* and *Cacajao*) were excluded from this analysis because only a few species of these taxa were available and this could affect the estimates of their probable ancestral areas.

In addition, we used a script to collapse the terminal taxa of the divergence time tree to represent the five species/monophyletic groups: *C. albinaus*, *C. satanas*, *C. utahicki*, *C. sagulatus* and *C. chiropotes*. This script is described in the section “How (and whether) to collapse tips to prune a tree” on the site: http://phylo.wikidot.com/example-biogeobearsscripts, accessed on 1 May 2022. Finally, we statistically compared the 12 different models via a comparison of LogLikelihood (ln L), Akaike Information Criterion (AIC, AICc) and ΔAIC. The Dispersion values (d), Extinction (e), Founder (J), Log-Likelihood criteria (ln L) and Akaike Information Criterion (AIC) were obtained from BioGeoBEARS. Three independent runs were conducted to confirm the reliability of the present results.

## 3. Results

The seven markers analyzed in the present study generated a database of 5694 base pairs (bp) per individual, with 1907 bp for mtDNA and 3787 bp for nDNA sequences, comprising 86 and 140 parsimony informative sites for mitochondrial and nuclear markers, respectively. Only 3.6% of missing data were observed.

The ML and BI analyses of all datasets recovered the same topologies, with five main clades in *Chiropotes*, placing *Cacajao* as the sister genus, and *Pithecia* and *Plecturocebus* as outgroups (Figure 2). The earliest lineage divergence of the extant *Chiropotes* taxa separated the *C. albinasus* clade from the ancestor of the other species. The subsequent divergence resulted in two clades, one containing *C. satanas* and *C. utahicki*, and another containing *C. chiropotes* and *C. sagulatus*. The BI analysis recovered maximum posterior probability values for all nodes, while the ML analysis recovered bootstrap values between 95% and 100%.

Based only on nuclear markers, the BI analysis resulted in a phylogeny with only three well-supported clades within *Chiropotes*: *C. albinasus*, *C. chiropotes-sagulatus* and *C. satanas-utahicki*. Note that *C. chiropotes* and *C. sagulatus* were recovered in distinct clades, but with no statistical support (pp = 0.62; Appendix A). However, BI from mtDNA markers (COI and Cytb b) recovered, significantly, five major clades: *C. albinasus*, *C. chiropotes*, *C. sagulatus*, *C. satanas* and *C. utahicki* (Appendix A).

The genetic distances between the main *Chiropotes* clades are presented in Table 2. Thus, the largest mean genetic distance was found between *C. albinasus* and the other species (7.0% for mtDNA and 3.0% for the concatenated markers). The mean distance between *C. satanas-utahicki* and *C. chiropotes-sagulatus* clades was 2.3% for mtDNA and 1.3% for the concatenated dataset. The distance between *C. satanas* and *C. utahicki* was 1.3% for mtDNA and 0.5% for the concatenated dataset, while the distance between *C. chiropotes* and *C. sagulatus* was 1.5% for mtDNA and 0.9% for the concatenated markers.

Our estimates of divergence times indicate that Callicebinae and Pitheciinae diverged about 16.34 Ma (95% HPD: 17.3–15.3 Ma), and that the most recent common ancestor of the pitheciines lived at about 9.56 Ma (95% HPD: 11.27–7.74 Ma) (Figure 3 and Table 3). The separation between *Chiropotes* and *Cacajao* occurred at approximately 5.87 Ma (95% HPD: 7.82–3.95 Ma). According to our estimates, *Chiropotes* diversified during the Plio-Pleistocene. *Chiropotes albinasus* was the first lineage to diverge, at approximately 4.18 Ma.

Ma (95% HPD: 6.12–2.33 Ma). The second divergence event occurred at 2.65 Ma (95% HPD: 4.65–1.18 Ma), when the lineage from the northern Amazon River (putative ancestor of *C. chiropotes* and *C. sagulatus*) diverged from the ancestors of *C. satanas* and *C. utahicki* (southern Amazon). Subsequent cladogenetic events gave rise to *C. satanas* and *C. utahicki* at 1.43 Ma (95% HPD: 3.1–0.34 Ma), and *C. chiropotes* and *C. sagulatus* at 1.39 Ma (95% HPD: 3.09–0.36 Ma).

We also analyzed the hypothesis of five valid species within *Chiropotes* using BPP and STACEY. The posterior probability values obtained in BPP were 0.85 for the node that connected *C. chiropotes* and *C. sagulatus* and 0.98 for the node connecting *C. satanas* and *C. utahicki*, while the remaining nodes were supported by a posterior probability equal to 1. STACEY presented maximum values of posterior probability for all nodes in the phylogeny (Figure 4). The DIVALIKE+J model produced the best statistical (AIC = 18.38 [ΔAIC = 0]) fit to the data in relation to the 12 biogeographic models evaluated in BioGeoBEARS (Appendix A). We verified that the DIVALIKE + J model explains the distribution of the current cuxius species based on four dispersal jump events (+J) (Figure 5).

The biogeographic analysis using BioGeoBEARS supported the hypothesis that the ancestor of the extant cuxius occupied the current areas of Rondônia and Tapajós endemism during the Early Pliocene. In these areas, the lineage that gave origin to *C. albinasus* was established. The DIVALIKE + J model assumes that the ancestral lineages of the other species reached the Belém endemism area by jump dispersal (+J). However, when we look at the pie chart in DIVALIKE + J, the Belém area shares similar probability values with three other areas of endemism (Xingu, Guyana and Tepui). Thus, our data are not sufficiently robust to recover the distribution area of the ancestors from *C. satanas*, *C. utahicki*, *C. chiropotes* and *C. sagulatus*.

Furthermore, DIVALIKE+J supported a second jump dispersal event in the Pliocene thatyes separated the lineages from the northern and southern Amazon River. The northern lineage was established in the current Guyana endemism area. Later, in the Pleistocene, the populations that occupied the areas of endemism in Guyana and Belém dispersed to Tepui and Xingu, respectively. This way, four lineages became genetically isolated in Belém, Xingu, Guyana and Tepui endemism areas, thereby giving rise to the species *C. satanas*, *C. utahicki*, *C. chiropotes* and *C. sagulatus.*

## 4. Discussion

### 4.1. Taxonomy and Phylogenetic Relationships of Chiropotes

The first insights into the taxonomy of the cuxius were derived from morphological and cytogenetic data [5,6]. Our findings, based on the analyses of DNA sequences, corroborate the early phylogenetic inferences [5,6], which identified *C. albinasus* as the most distinct cuxiu species. The lineage that originated *C. albinasus* diverged from the ancestor of all other cuxiu species during the Pliocene (~4.18 Ma). This time interval allowed the accumulation of morphological and behavioral differences, such as the skull size and the peculiar configuration of beard and tail, vocalization and social behavior, that clearly differentiate *C. albinasus* from the remaining species [6].

The results from ML and BI analyses recovered four closely related taxa of cuxius, except for *C. albinasus*. Based on divergence time estimates, the common ancestor of these four taxa existed nearly 2.65 Ma, which corresponds to the transition between the Pliocene and Pleistocene.

Subsequently, two more diversification events within *Chiropotes* took place ~1 Ma (see Figure 5), thus giving rise to the lineages related to *C. satanas*, *C. utahicki*, *C. chiropotes* and *C. sagulatus*. Based on this result, we applied two species-level analyses (BPP and STACEY) to verify whether all four clades are valid species. Both STACEY and BPP rely on the multispecies coalescent model to estimate the species delimitation. However, STACEY uses a prior birth–death–collapse model without the need for reversible-jump Markov Chain Monte Carlo (rjMCMC) and a guide tree [39]. On the other hand, species delimitation in BPP is based on rjMCMC from a guide tree and the proper definition of prior population size and time of divergence. Our results regarding species delimitation performed in STACEY suggest that *C. satanas*, *C. utahicki*, *C. chiropotes* and *C. sagulatus* are full species, since all nodes in the cladogram were recovered with maximum posterior probability values. On the other hand, BPP does not fully support that *C. chiropotes* and *C. sagulatus* represent two species, while *C. satanas* and *C. utahicki* were recovered as distinct species.

The validation of *C. satanas* as a full species has important implications for the conservation of this cuxiu lineage since it occurs along the eastern Tocantins-Araguaia basin within the Belém center of endemism (BE). Because of the long history of human impacts, the BE is the most degraded portion from the arc of deforestation in the southern Amazon, mainly related to the conversion of native forests into cattle pastures and soybean and eucalypt crops. Nowadays, only 24% of the original forest cover remains within the BE [47], thus causing major biodiversity losses and increasing the threats to local species, including *C. satanas*, currently classified as critically endangered [48].

The species status of the clades from the eastern and western Branco River (herein referred to as *C. sagulatus* and *C. chiropotes*, respectively) was conflicting. While the analysis in STACEY recovered both groups as full species, the BPP results were not consistent. This discrepancy is likely to be associated with the small sample size of both lineages. Alternatively, these putative taxa that inhabit opposite banks of the Branco River might represent not fully isolated evolutionary lineages. However, other sources of evidence, such as the differences in fur coloration and the presence of distinctive pericentric inversions in karyotypes of each lineage from opposite banks of the Branco River, suggest that these taxa are distinct species [8]. Therefore, taking into account the sum of molecular, chromosomal [6] and morphological [5] data and following the premises of the phylogenetic species concept, we infer that *C. chiropotes* and *C. sagulatus* should be regarded as two valid species.

### 4.2. Diversification and Biogeographic History of Chiropotes

The lineage of the extant *Chiropotes* diversified ~4 Ma, originating *C. albinasus* and the ancestor of the other species. After splitting, *C. albinasus* was distributed throughout Rondônia and Tapajós centers of endemism. However, the present biogeographic analyses were inconclusive about the distribution area of the ancestors from the remaining species of cuxiu. The *C. satanas*/*C. utahicki* clade is found south of the Amazon River, along Xingu and Belém centers of endemism, whereas the *C. chiropotes*/*C. sagulata* clade occurs to the north of the Amazon River. Even though the area occupied by the ancestor of both clades could not be reliably determined, the analysis in BioGeoBEARS suggested the Belém area as the most likely ancestral area.

Considering these uncertainties, we infer two plausible scenarios: (i) the ancestor of the *C. satanas* clade was widely distributed and the formation of the Amazon River acted as a barrier that separated northern and southern lineages; (ii) the ancestor of the *C. satanas* clade originally inhabited one margin of the Amazon River and, later, a population dispersed to the opposite margin. In spite of the controversies about the timing of the formation of the Amazon River [45,49], all reported estimates are before the period we inferred for the divergence between northern and southern clades of the *C. satanas* lineage (about 2.65 Ma). Therefore, the formation of the Amazon River apparently played no significant role in the separation of these cuxiu clades. By contrast, the transposition of the Amazon River should represent the most probable scenario, as proposed for other Amazonian primates, such as *Saimiri* [50] and *Saguinus* [51]. In this case, the ancestor population of the *C. satanas* clade from one margin of the Amazon River eventually crossed this river at around 2.65 Ma and dispersed towards the opposite bank. Nonetheless, it should be pointed out that the Amazon is one of the largest rivers in the world, reaching at least a width of 2 km in the narrowest point (in Óbidos, state of Pará) on the eastern half of the basin [52], which minimizes the possibility of active crosses between margins by these primates. However, during the glacial maxima in the Pleistocene, the water volume of the Amazon River drastically decreased [53], thus favoring their dispersal between opposite margins.

It is also possible that a passive transposition occurred, given that the course of the Amazon River has constantly shifted over geological times [54]. These shifts with the formation of oxbow lakes in particular may have isolated populations from one river margin to another. Assuming that *Chiropotes* first appeared in the Rondônia-Tapajós endemism areas, i.e., the southern Amazon River, the most parsimonious interpretation would be the initial occurrence of the ancestors of the other species (*C. satanas*, *C. utahicki*, *C. chiropotes*, and *C. sagulatus*) throughout the southern region of the Amazon River basin, followed by the population transposition to the northern margin. Any other interpretation would require the occurrence of two transposition events during different periods.

Around one million years ago, the populations that occupied both Belém and Guyana endemism areas dispersed westwards and the migrants established themselves in the Xingu and Tepui endemic areas, respectively, following a founder–event speciation model. Geological evidence indicates that the Branco River and the Tocantins-Araguaia basin were formed at about 1 Ma [55,56]. In this scenario, we believe that the formation of the Branco River isolated two lineages in the Tepui and Guyana areas along the northern Amazon River, while the formation of the Tocantins-Araguaia basin restricted the gene flow between the lineages that occurred in the current Belém and Xingu endemism areas. These four geographically and genetically isolated lineages over the past 1 million years gave rise to *C. satanas* in Belém, *C. utahicki* in Xingu, *C. sagulatus* in Guyana and *C. chiropotes* in Tepui.

## 5. Conclusions

The present study provides the first taxonomic hypothesis for *Chiropotes* based on a multi-loci approach, using both nuclear and mitochondrial markers. Our data reinforce the taxonomy validity of five cuxiu species (*C. albinasus*, *C. satanas*, *C. utahicki*, *C. chiropotes* and *C. sagulatus*), placing *Chiropotes albinasus* as the most genetically divergent taxon. The large genetic distance of this species compared to the others is related to the early split of the lineage that originated *C. albinasus* from the remaining groups in the Early Pliocene (~4 Ma). Such genetic differentiation of *C. albinasus* also reflects the unique phenotypic traits of this species in relation to other cuxiu representatives. Because of these external differences, Hershkovitz [6] recognized *C. albinasus* as a valid species and proposed that *C. satanas* should comprise three subspecies. The subspecies suggested by Hershkovitz [6] diversified later in the Pleistocene and indeed show little genetic and morphological differences among them. However, we endorse the recognition of five *Chiropotes* species, since they represent reciprocally monophyletic lineages with ranges separated by geographic barriers that have prevented gene flow for approximately 1 Ma. In fact, the correct taxonomic assessment of cuxius is a key feature of their conservation inasmuch as species represent the smallest evolutionary units used in conservation management and policies. Accordingly, *C. satanas*, as defined in the present study, is a critically endangered species [13] found only in the Belém endemism area.

This study also provides the first biogeographic hypothesis for the origin of cuxius, indicating that the ancestors of *Chiropotes* inhabited the southern region of the Amazon River basin in the present Rondônia and Tapajós endemism areas. From these areas, a population radiated towards the eastern Amazon, while the primates that remained in the Rondônia-Tapajós area gave rise to *C. albinasus* in the Late Pliocene. The lineage that migrated eastwards was established in Belém, and later, some individuals dispersed to the northern Amazon, more precisely to the Guyana endemism area. During the Pleistocene, two additional migratory events took place (from Guyana to Tepui and from Belém to Xingu). These events were accompanied by the formation of the Branco River and the Tocantins-Araguaia Basin, thus isolating the populations from Xingu and Tepui endemism areas, respectively, and giving rise to the extant five species of cuxius.

## Figures and Tables

**Figure 1 genes-14-01309-f001:**
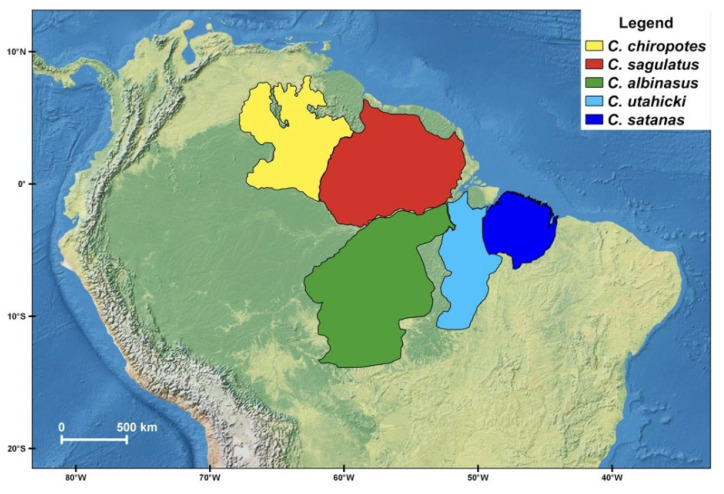
Geographic distribution of the extant species of the genus *Chiropotes* based on Hershkoviz [6,7,8].

**Figure 2 genes-14-01309-f002:**
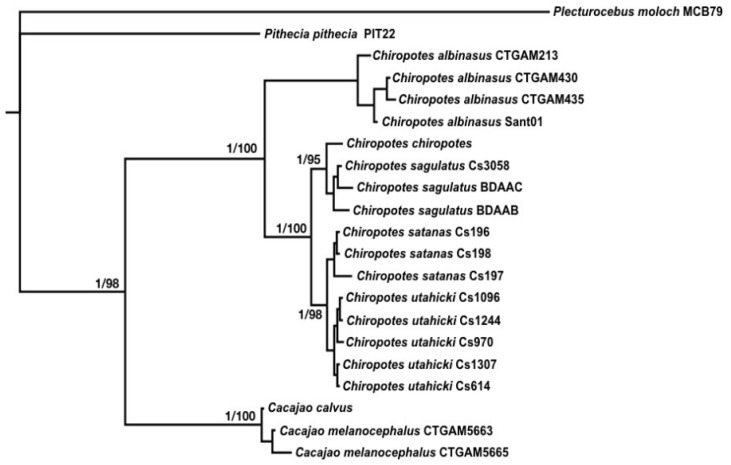
Phylogenetic reconstruction by Maximum Likelihood (ML) and Bayesian Inference (BI) approaches. The numbers adjacent to the nodes represent the posterior probabilities (PP) of BI and percentage bootstrap values for the ML analyses. Nodes without numbers have BI = 1 and ML = 100%.

**Figure 3 genes-14-01309-f003:**
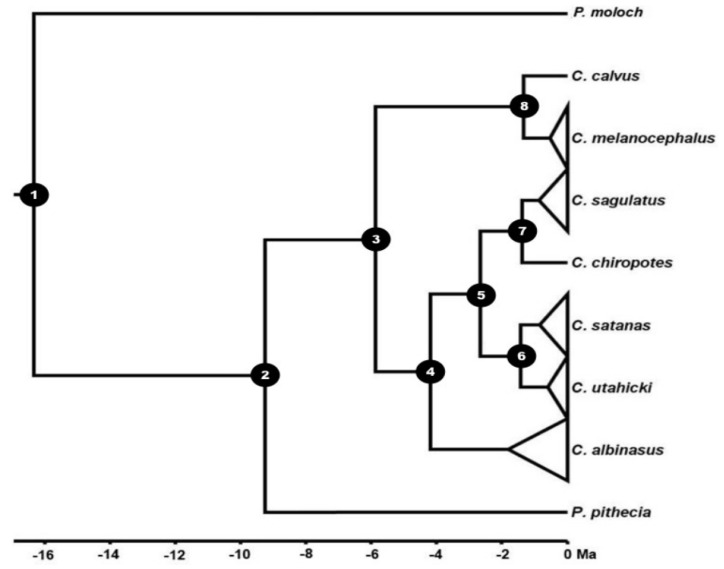
Estimates of divergence times among taxa of the family Pitheciidae. Values in the abscissa indicate the time of separation between clades. The 95% HPD values are shown in Appendix A.

**Figure 4 genes-14-01309-f004:**
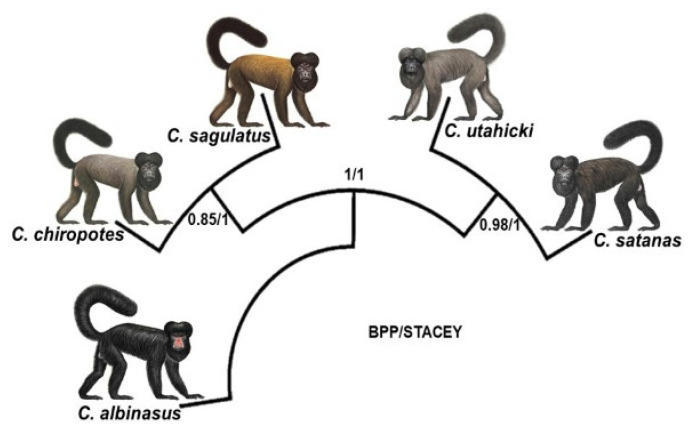
Species delimitation analyses performed using Bayesian Phylogenetics and Phylogeographic (BPP) and Species Tree and Classification Estimation, Yarely (STACEY). Numbers at nodes indicate the posterior probabilities.

**Figure 5 genes-14-01309-f005:**
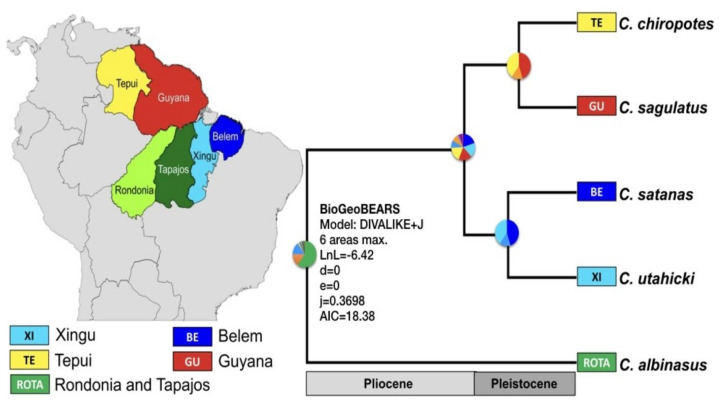
Time-calibrated species trees generated by BEAST and ancestral range estimates provided by BioGeoBEARS. The best-fitting model found for the ancestral range estimation was DIVALIKE + J (lnL = −6.42 AIC = 18.38). Node pie charts represent the likelihoods of ancestral states; the color-coded circles at the tips represent the current areas occupied by each lineage.

**Table 1 genes-14-01309-t001:** Species, specimen codes and institution in which the DNA samples are stored and the geographical origins of each specimen.

Species	Code	Institution	Locality
*Chiropotes satanas*	Cs196	UFPA	1
*Chiropotes satanas*	Cs197	UFPA	1
*Chiropotes satanas*	Cs198	UFPA	1
*Chiropotes utahicki*	Cs1096	UFPA	2
*Chiropotes utahicki*	Cs1307	UFPA	2
*Chiropotes utahicki*	Cs970	UFPA	2
*Chiropotes utahicki*	Cs1244	UFPA	2
*Chiropotes utahicki*	Cs614	UFPA	2
*Chiropotes albinasus*	CTGAM213	UFAM	3
*Chiropotes albinasus*	CTGAM430	UFAM	3 *
*Chiropotes albinasus*	CTGAM435	UFAM	3 *
*Chiropotes albinasus*	Sant01	UFPA	4
*Chiropotes sagulatus*	Cs3058	UFPA	5
*Chiropotes sagulatus*	BDAAC	UFPA	5
*Chiropotes sagulatus*	BDAAB	UFPA	5
*Chiropotes chiropotes*	No code	GenBank	6
*Cacajao calvus*	No code	GenBank	NI
*Cacajao melanocephalus*	CTGAM5663	UFAM	7
*Cacajao melanocephalus*	CTGAM5665	UFAM	7
*Pithecia Pithecia*	Pit22	UFPA	8
*Plecturocebus moloch*	MCB79	UFPA	9

1: east bank of the Tocantins River, UHE Tucuruí, Pará, Brazil; 2: west bank of the Tocantins River, UHE Tucuruí, Pará, Brazil; 3: west bank of Tapajós River; 3 *: east bank of Tapajós River; 4: east bank of the Tapajós River, Santarém, Pará, Brazil; 5: west bank of the Trombetas River, Cachoeira Porteira, Oriximiná, Pará, Brazil; 6: east bank of the Mucajaí River, Roraima, Brazil; 7: east bank of Negro River, Amazonas, Brazil; 8: east bank of the Jari River, Amapá, Brazil; 9: west bank of the Xingu River, UHE Belo Monte, Pará, Brazil; NI; No information.

**Table 2 genes-14-01309-t002:** Pairwise genetic distances among *Chiropotes* taxa. The percentages in gray refer to the mtDNA marker while those in orange refer to the concatenated sequences.

	*C. albinasus*		*C. satanas*		*C. utahicki*		*C. sagulatus*	
*C. satanas*	6.8%	2.7%						
*C. utahicki*	6.9%	2.7%	1.3%	0.5%				
*C. sagulatus*	7.4%	2.8%	2.3%	1.5%	2.3%	1.5%		
*C. chiropotes*	7.0%	2.7%	2.5%	1.3%	2.4%	1.3%	1.5%	0.9%

**Table 3 genes-14-01309-t003:** Divergence date estimates and 95% highest posterior densities for each node of the cladogram shown in Figure 3.

Node	Clade	Median Age in Ma	95%HPD	
1	Callicebinae vs. Pitheciinae	16.34	17.3	15.3
2	*Pithecia* vs. *Cacajao* + *Chiropotes*	9.56	11.27	7.74
3	*Cacajao* vs. *Chiropotes*	5.87	7.82	3.95
4	*C. albinasus* vs. *satanas* clade	4.18	6.12	2.33
5	*C. chiropotes* + *C. sagulatus* vs. *C. satanas* + *C. utahicki*	2.65	4.65	1.18
6	*C. satanas* vs. *C. utahicki*	1.43	3.1	0.34
7	*C. chiropotes* vs. *C. sagulatus*	1.39	3.09	0.36
8	*Cacajao calvus* vs. *Cacajao melanocephalus*	1.34	1.87	0.02

## Data Availability

The aligned sequences, script for BioGeoBears and Control file for BPP analysis associated with the genes in this study are available in Dryad: https://doi.org/10.5061/dryad.mgqnk98wb.

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
