# Peer review of "Molecular Evidence Supports Five Lineages within Chiropotes (Pitheciidae, Platyrrhini)"

_genes, 2023, doi:10.3390/genes14071309_

Round 1

Reviewer 1 Report

While most of the research work is good, some aspects of the manuscript are not well written, as the work needs to improve its english and some mistakes such as genera were not written in italics or the species author names in brackets or not (as required) which are basic for a taxonomic study, were not double checked.    

While most of the research work is good, some aspects of the manuscript are not well written, as the work needs to improve its english and some mistakes such as genera were not written in italics or the species author names in brackets or not (as required) which are basic for a taxonomic study, were not double checked.    

Author Response

Virtually all of the reviewers' suggestions were accepted. Except two:

Point 1. User a better introduction is required here. It has to be more scientific Point 2. Place icons on the map to represent the collection points.

Response 1:  I don't understand why the beginning of the introduction is not suitable for scientific writing

Response 2: Below table 1 that refers to the details of the samples, we inform the location where each sample was collected, we think it would be redundant to include the same information on the map from icons

Reviewer 2 Report

The manuscript « «  by Carneiro et al. presents a phylogenetic analysis of the genus Chiropotes using 16 specimens and 2 mt genes and 5 nuclear regions.

The authors used uptodate tools for phylogeny construction and monophyly inference.

There are several points that require the attention of the authors.

First, such an analysis cannot be used to conclude about species delimitation; it can be used to conclude about monophyletic lineages. The conversion from monophyletic lineage into species is completely subjective. “systematic/phylogenetics is the ‘hard’ science and taxonomy as the art of interpreting that science” See for example this discussion in:

Pisupati B (2015) Taxonomy - the science and art of species. Current Science 108: 2149-2150

For this reason, the title of the paper should be rather :

Molecular evidences support five lineages within Chiropotes (Pitheciidae, Platyrrhini)

This is particularly true with the data presented by the authors because only one sample come from Chiropotes chiropotes species - this sample is labelled Chiropotes israelita in genbank based on 

Perelman P, Johnson WE, Roos C, Seuanez HN, Horvath JE, Moreira MA, Kessing B, Pontius J, Roelke M, Rumpler Y, Schneider MP, Silva A, O'Brien SJ, Pecon-Slattery J (2011) A molecular phylogeny of living primates. PloS Genetics 7: e1001342 doi 10.1371/journal.pgen.1001342

As the sampling of individuals is low, it is important to locate them with crosses in the map Figure 1.

The authors should discuss the paucity of the samples used in the study: only 16 specimens for 5 lineages (putative species) and approximately 4 000 000 km2!

Minor changes :

Guyana (line 212) should be rather named Guianas because Guyana is a country and Guianas is a region.

Confirm that C. chiropotes in UICN status (line 59) comprised both C. chiropotes and C. sagulatus.

Author Response

We agree with the reviewer and have changed our title and information in the manuscript to indicate that our results suggest five lineages and not valid species as previously. The other minor suggestions were also accepted
